

# The impact of chemical and hormonal treatments to improve seed germination and seedling growth of *Juniperus procera* Hochst. ex Endi

Alae Ahmad Jabbour[1] and Abdulaziz Alzahrani[2]

[1] Department of Biology, Faculty of Applied Science, Umm Al-Qura University, Makkah, Saudi Arabia
[2] Department of Biology, Faculty of Science, Al-Baha University, Alaqiq, Al-Baha, Saudi Arabia

Corresponding author
Alae Ahmad Jabbour,
alaeperesearch@hotmail.com

## ABSTRACT

**Purpose**. Juniper (*Juniperus procera*) is a common forest tree species in Saudi Arabia. The decline in many populations of *J. procera* in Saudi Arabia is mainly due to seed dormancy and loss of natural regeneration. This study assessed the effects of chemical and hormonal treatments on seed germination and seedling growth in juniper plants. **Methods**. The seeds were subjected to either chemical scarification with 90% sulfuric acid and 20% acetic acid for 6 min or hormonal treatment by seed soaking in two concentrations (50 and 100 ppm) of three growth regulators, namely, indole acetic acid (IAA), gibberellins ($GA_3$), and kinetin, for 72 h. A control group without any seed treatment was also prepared. The experiments were performed in an incubator maintained at room temperature and under a light and dark period of 12 h for 6 w. The germinated seeds for each treatment were counted and removed from the dishes. The selected germinated seeds from different treatments were planted in a greenhouse and irrigated with tap water for another 6 weeks. The hormone-treated seedlings were sprayed with their corresponding hormone concentrations 1 w after planting. **Results**. The highest percentage of seed germination was significantly recorded after seed soaking in 50 ppm $GA_3$, whereas treatment with IAA (100 ppm) resulted in the best seedling growth. Seedlings treated with the three phytohormones showed a significant increase in photosynthetic pigments, total soluble sugars, proteins, percentage of oil, IAA, $GA_3$, and kinetin contents of juniper seedlings compared with the control value, whereas abscisic acid content was decreased compared with chemical treatments. **Conclusion**. The investigated different treatments had an effective role in breaking seed dormancy and improving seedling growth of *J. procera*, which is facing a notable decline in its population worldwide. Moreover, such an effect was more pronounced in the three phytohormones that succeeded in breaking dormancy and growth of the *Juniperus* plant than in the other treatments.

## INTRODUCTION

Through a wide distribution that covers most of Saudi Arabia, *Juniperus procera* L. is one of the most widely spread plant species on earth. Juniper forest is the most dominant

natural forest on the southwestern side of Saudi Arabia (*Khalofah et al., 2022*). *J. procera* is one of the most widely distributed conifers after *Taxus baccata* and *Pinus sylvestris*. It is an evergreen perennial gymnosperm shrub belonging to the family Cupressaceae with fleshy fruits. Juniper forest has many benefits because it prevents soil erosion, helps in ecological balance and grazing, and serves as a carbon storage (*Hernández & Clemente, 1994*; *Ahani et al., 2013*). Moreover, juniper trees are economically important because they provide firewood, construction materials, and beekeeping (*Abo-Hassan, El-Osta & Sabry, 1984*). Lead pencils and timber wood for buildings and outdoor structures are made from juniper trees (*Cantos et al., 1998*; *Mamo et al., 2011*).

In recent decades, various studies have reported declining numbers and sizes of juniper populations in different areas, particularly in Mediterranean mountainous areas (*García et al., 1999*). The decline in juniper forest is a result of several biotic and abiotic factors, including grazing and low natural regeneration capacity (*Hajar, Faragalla & Al-Ghamdi, 1991*; *Aref & El-Juhany, 2004*; *El-Juhany, 2009*).

Although grazing plays an efficient role and helps in the achievement of seed germination (*Hommel et al., 2009*; *Broome et al., 2017*), there are risks to vegetation triggered by grazing, which can affect new seedlings by eliminating and damaging mature shrubs (*Clifton, Ward & Ranner, 1997*; *Broome et al., 2017*). The grazing management type is considered a threatened status of the remaining juniper populations in the Saudi Arabia region; therefore, additional management forms are required to achieve suitable practices for seed germination and survival of juniper seedlings.

*J. procera* has been recorded as a threatened species because of overexploitation and global decline in its population (*Farjon, 2013*). The low rate of regular regeneration of *J. procera* reached up to 50% among trees in some parks, in addition to the absence of new regeneration (*Barth & Strunk, 2000*). Factors that threaten population decline have been attributed to climate change (*Fisher, 1997*), soil erosion and water run, decreasing annual rainfall (*Aref et al., 2013*; *El Atta et al., 2013*; *Hosny & Almazroui, 2015*), and human interference (*Gardner & Fisher, 1994*). Dieback have been reported in juniper ecosystems in the Arabian Peninsula, southwest Asia (*Fisher, 1997*), and parts of Africa (*Borghesio et al., 2004*).

Juniper seeds exhibit deep dormancy, which decreases their regeneration capacity (*Bonner et al., 2008*). The degree of dormancy is variable among species; juniper species recorded several germination percentages (*Bonner et al., 2008*).

Seed dormancy is defined as the failure of the seed to germinate and form a normal seedling under favorable conditions for germination, such as a humid substrate, adequate oxygen supply, and optimum temperature. Seed dormancy may be due to an impermeable seed coat (physical dormancy), an underdeveloped embryo (morphological dormancy), the presence of biochemical and physiological inhibitory mechanisms (physiological dormancy), or the integration of the three types (*Baskin & Baskin, 2004*). Many juniper species have physiological or morphophysiological dormancy (*García-Fayos et al., 2002*; *Al-Refai et al., 2003*; *Tilki, 2007*; *Tylkowski, 2009*; *Tylkowski, 2010*); however, studies on physical dormancy have provided contradictory data (*García-Fayos et al., 2002*; *Society for Ecological Restoration (SER), 2008*). Improving the seed germination process

and accelerating seedling growth of juniper are the most important steps in improving their regeneration capacity.

Seed dormancy with physiological or morphophysiological types of dormancy may be released by stratification with either cold or warm–cold, a combination of light and hormonal treatments (*Baskin & Baskin, 2001*; *El-Juhany, 2009*; *Afroze & O' Reilly, 2013*).

Germination is regulated by phytohormones, including auxins, gibberellic acid, abscisic acid (ABA), and ethylene (*Han & Yang, 2015*). Therefore, the objective of this study was to assess the effect of plant growth-regulating substances, 90% sulfuric acid and 20% acetic acid, on the seed dormancy and seedling growth of *J. procera*.

## MATERIALS AND METHODS

### Seed samples and collection processing

*J. procera* Hochst. ex Endi. cones were collected from Baljurashi, Al-Baha, Saudi Arabia (19°51′18.72″N, 41°33′33.29″E). Cones were collected from at least 10 randomly selected mother trees. The collected cones were transported to the laboratory within 4 to 5 days after collection, where the seeds were extracted by manual depulping, cleaned, and air dried to approximately 5%–8% moisture content. The air-dried seeds were stored in the refrigerator until use.

### Imposition of treatments and germination experiment

*J. procera* seeds were superficially sterilized with sodium hypochlorite (1%) for approximately 5 min, then rinsed carefully with distilled water, and divided into nine groups. The first group soaked in distilled water served as the control group. The second and third groups were soaked in 90% sulfuric acid and 20% acetic acid, respectively, for 6 min and then washed many times with tap water. The other six groups were soaked in either 50 or 100 ppm indole acetic acid (IAA), $GA_3$, and kinetin for 72 h. Three replicates of 20 seeds from each treatment were placed into 11-mm Petri dishes on sterilized filter paper that was kept continuously moist with distilled water. All Petri dishes for all treatments of the experiment were placed in an incubator that was kept at room temperature $24\,°C \pm 1\,°C$ with alternating light and dark conditions. The germination experiment was continued for 6 weeks, and when germinated seeds formed 3-mm radicles, the germination percentage for all treatments was calculated. The germinated seeds from each treatment were removed from the Petri dishes for the seedling growth experiment.

### Seedling growth experiment

Germinated seeds from each treatment were planted in 10-cm plastic pots filled with a homogenate of sterilized peat moss and sand at a ratio of (1:1, v/v). All were retained in a greenhouse with a randomized complete block design with three replicates for each treatment and were irrigated with distilled water for 6 weeks. Two weeks after planting, each hormone-treated seedling was sprayed with its own hormone concentration.

### Methods
#### Seed germination and growth criteria

(Seed germination % = number of germinated seeds/total number of seeds $\times$ 100)

After the end of the seedling growth experiment, 10 replicates of plants were randomly sampled from each treatment, cautiously washed with distilled water, and left on the filter sheets to remove water for measurement of growth criteria. Plant growth parameters were recorded, *i.e.,* lengths of roots and shoots and fresh and dry weights of seedlings. The dry weight of seedlings was measured in an oven at 70 °C until persistent dry weights were observed.

### Extraction and estimation of photosynthetic pigments

Photosynthetic pigments (chlorophylls a and b and carotenoids) were extracted and estimated using the method adopted by *Metzner, Rau & Senger (1965)*. A known fresh weight of leaves was mixed in 85% acetone for approximately 5 min. The homogenate was filtered using Whatman filter paper no. 1, and the filtrate was made up to a known volume using 85% acetone. Chlorophylls a and b and carotenoids were measured at wavelengths of 663, 644, and 452.5 nm, respectively.

### Extraction and estimation of carbohydrates

A commonly used method for the extraction of carbohydrates was described by *Homme, Gonzalez & Billard (1992)*. The method used for the determination of soluble sugars was that of *Blakeney & Mutton (1980)*. Approximately two mL of plant extract or standard solution and an adequate amount (10 mL) of anthrone (0.1 g anthrone in 72% $H_2SO_4$) were added, and all tubes were boiled in a water bath for 20 min. After cooling, the absorbance was measured at 620 nm using a spectrophotometer.

### Extraction and estimation of total soluble proteins

Total soluble proteins were determined using Folin–Ciocalteu reagent as described by *Daughaday et al. (1952)*. The plant protein sample (0.1 mL) was added to five mL of alkaline reagent in a clean test tube. The tubes were allowed to stand at room temperature for at least 15 min. Then, 0.5 mL of the diluted Folin–Ciocalteu reagent (1:2 v/v) was added to the aforementioned mixture and immediately mixed. After 30 min, absorbance was measured at 700 nm using a spectrophotometer.

### Extraction and determination of the oil content

The method used for extracting oil was similar to that explained by *AOAC (2002)*. A known weight of juniper (*J. procera*) seedlings was ground in a mortar and then transferred to a fat-free extraction thimble, and extraction with hexane in a soxhlet apparatus was sustained for 10 h. The resultant oil extract was then transferred to a weighed flask. The solvent was evaporated in air. The last remaining drops of the solvent were evaporated by heating at 100 °C in an oven; after cooling the flask in a desiccator, the flask was weighed again. The difference in weight was equivalent to the weight of the oil content in the plant sample.

### Extraction and estimation of endogenous phytohormones

Endogenous phytohormones were assayed following the method described by *Shindy & Smith (1975)*. The injection of 10 μL into HPLC 510b was adjusted for the identification and estimation of hormones using a data model (Waters 746), a detector (UV tunable absorbance), and a pump (HPLC 510). The chromatograph was fitted (equipped with a

$3.9 \times 300$ mm Bondapak C18 capillary column). HPLC was run at 25 °C. Retention time, peak area of various phytohormones, and authentic standards were used to identify and determine sample peaks.

## Statistical analysis

Data are expressed as mean values ± standard error. Statistical analysis was performed using the one-way analysis of variance test followed by Dunkin's test accompanied by the least significant difference test at $P$ value < 0.05 (*Snedecor & Cochran, 1990*).

## RESULTS AND DISCUSSION

### Effect of growth regulators on seed germination and seedling growth in *Juniperus procera*

Recently, numerous techniques have been performed to achieve seed priming, such as hydropriming, osmopriming, and thermopriming (*Ashraf & Foolad, 2005*). The treatments could have different effects according to the priming agent level/dose and its incubation period and depending on the species and stage of plant development. Moreover, various studies on seed germination have shown the useful effects of seed priming using many techniques such as soaking, temperature, and scarification (*Ahmed et al., 2006*).

Exogenous application of growth regulators improved the seed germination process and seedling growth of plants, which induced various physiological responses (*Ries, 1991*; *Terzi & Kocaçalışkan, 2010*). It plays an effective role in the upregulation of various biochemical and physiological aspects in plants (*Ivanov & Angelov, 1997*; *Chen et al., 2003*).

The results obtained in this study indicate a significant change in the main growth attributes and physiological activities of the developed *J. procera* seedlings in response to soaking grains in either IAA, gibberellin ($GA_3$), or kinetin with relatively low (50 ppm) or high (100 ppm) concentrations. Thus, in response to soaking *J. procera* seeds in either relatively low or high concentrations, the emergence of radicles was noticeable. This was accompanied by a concomitant significant increase in the germination percentage, lengths of shoots and roots, fresh and dry weights of seedlings, and leaf contents of chlorophylls a and b and carotenoids (Fig. 1 and Tables 1 and 2).

Among all treatments used in this study, when juniper seeds were treated with 50 ppm $GA_3$, a maximum percentage germination (40%) was recorded, followed by 100 ppm $GA_3$ and 50 ppm kinetin, which recorded 30% of germination. Then, 100 ppm kinetin and 50 and 100 ppm IAA showed the lowest percentage of germination as growth regulators. However, the juniper seeds that were chemically scarified with either 90% sulfuric acid or 20% acetic acid significantly recorded a minimum percentage of germination (10%) over the control value ($H_2O$), which recorded 0% germinated seeds (Fig. 1). Our results are in accordance with those of *Banerji (1998)*, who reported that the response of $GA_3$ had a better performance than IAA on the germination of *Melia azedarach* seeds.

The highest percentage of germination recorded with $GA_3$ treatments might be attributed to $GA_3$ acting on the embryo and induced the synthesis of hydrolytic enzymes involved in seed germination especially $\alpha$-amylase and protease, which consequently hydrolyzed seed storage substance reserves such as insoluble sugar and proteins into

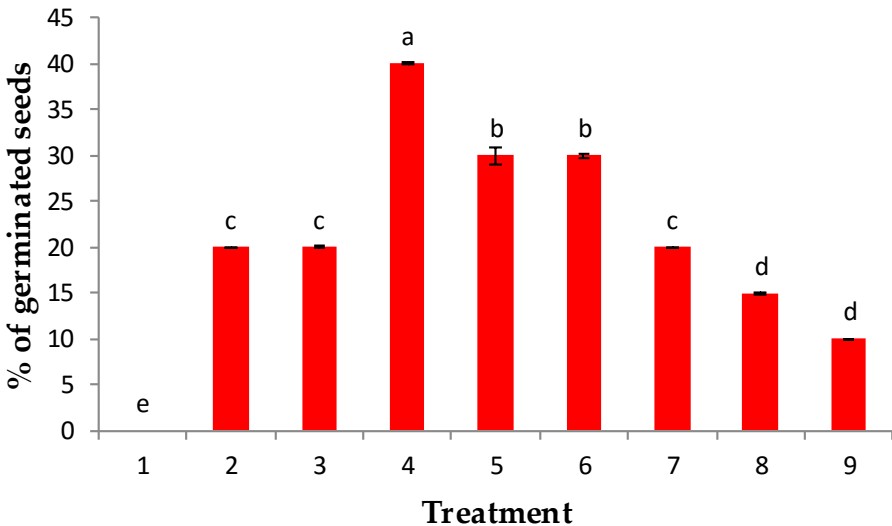

**Figure 1** Effect of pre-treatment with (1) control, (2) IAA at 50 ppm, (3) IAA at 100 ppm, (4) GA₃ at 50 ppm, (5) GA₃ at 100 ppm, (6) kinetin at 50 ppm (7) kinetin at 100 ppm, (8) 90% sulphuric acid and (9) 20% acetic acids on % of germination of the percentage of germination of *Juniper procera* seeds.

**Table 1** Effect of pre-treatment with IAA at 50 ppm, IAA at 100 ppm, GA₃ at 50 ppm, GA₃ at 100 ppm, kinetin at 50 ppm kinetin at 100 ppm, 90% sulphuric acid and 20% acetic acids on shoot length, root length, fresh and dry weights of *juniper procera* seedling. Data are means of three replications ± SE. Each value is the mean of ten replicates ± SE.

| Treatment | Shoot length (cm) | Root length (cm) | Seedling fresh weight (g) | Seedling dry weight (g) |
|---|---|---|---|---|
| IAA (50 ppm) | 3.4 ± 0.5[a] | 6.7 ± 0.1[b] | 0.17 ± 0.01a[b] | 0.03 ± 0.009[b] |
| IAA (100 ppm) | 3.6 ± 0.06[a] | 7.6 ± 0.23[a] | 0.19 ± 0.005[a] | 0.05 ± 0.012[a] |
| GA3 (50 ppm) | 3.1 ± 0.05[b] | 5.6 ± 0.12[c] | 0.15 ± 0.006[bc] | 0.07 ± 0.003[a] |
| GA3 (100 ppm) | 2.9 ± 0.07b[c] | 5.2 ± 0.05[c] | 0.1 ± 0.005[c] | 0.06 ± 0.012[a] |
| Kinetin (50 ppm) | 2.9 ± 0.1b[c] | 4.5 ± 0.3[c] | 0.13 ± 0.006[c] | 0.021 ± 0.006[b] |
| Kinetin (100 ppm) | 2.8 ± 0.17[c] | 3.8 ± 0.17[e] | 0.12 ± 0.01c[c] | 0.02 ± 0.005[b] |
| Sulphuric acid (90%) | 2.1 ± 0.05[c] | 3.1 ± 0.06[f] | 0.1 ± 0.017[c] | 0.01 ± 0.004[b] |
| Acetic acids (20%) | 2.0 ± 0.0[c] | 3.2 ± 0.3[f] | 0.11 ± 0.005[c] | 0.012 ± 0.00[b] |

**Notes.**
Columns with different lowercase letters are significantly different at $p < 0.05$.

simple mobilizing soluble sugar and proteins, which are consumed for embryo growth and consequently enriched germination and expansion processes (*Paleg, 1965*; *Bewley & Black, 1994*; *Kaneko et al., 2002*). Moreover, the growth regulators GA₃ and IAA could inhibit these inhibitors because ethylene exists in seeds. Similar findings were confirmed

**Table 2 Effect of pre-treatment with IAA at 50 ppm, IAA at 100 ppm, GA$_3$ at 50 ppm, GA3 at 2 100 ppm, kinetin at 50 ppm kinetin at 100 ppm, 90% sulphuric acid and 20% acetic acids on three photo-synthetic pigments of *juniper procera* leaves.** Data are means of three replications ± SE. Each value is the mean of three replicates ± SE.

| Treatment | Chlorophyll a (μg/g F.wt) | Chlorophyll b (μg/g F.wt) | Carotenoids (μg/g F.wt) |
|---|---|---|---|
| IAA (50 ppm) | 285.1 ± 2.9[c] | 98.5 ± 1.2[b] | 125.1 ± 1.1[c] |
| IAA (100 ppm) | 314.4 ± 0.23[c] | 98.9 ± 0.4[b] | 159.3 ± 0.17[c] |
| GA3 (50 ppm) | 401.6 ± 0.6[a] | 121.6 ± 1.7[a] | 198.5 ± 1.2[a] |
| GA3 (100 ppm) | 380.5 ± 0.3[b] | 118.7 ± 0.12[a] | 179.7 ± 0.6[b] |
| Kinetin (50 ppm) | 268.4 ± 1.1[e] | 93.9 ± 1.7[c] | 150.3 ± 0.57[c] |
| Kinetin (100 ppm) | 237.2 ± 1.2[f] | 72.4 ± 1.1[c] | 155.8 ± 2.8[c] |
| Sulphuric acid (90%) | 197.7 ± 1.7[g] | 61.8 ± 0.2[e] | 114.6 ± 1.7[e] |
| Acetic acids (20%) | 172.1 ± 1.1[h] | 45.3 ± 2.8[f] | 100.8 ± 0.12[f] |

**Notes.**

Columns with different lowercase letters are significantly different at $p < 0.05$.

by *Suryakanth, Mukunda & Raghavendraprasad (2005)* and *Babu et al. (2010)* in guava and papaya plants, respectively. Furthermore, the soaking of juniper seeds in either sulfuric or acetic acid for approximately 6 min stimulated the percentage of germination above the control value; water treatment (0%), but still the lowest values of germination compared with the investigated growth regulators (Fig. 1). The reduction in germination percentage in both acids may be due to the toxic influence of ions present in the acids, particularly sulfate; moreover, these acids may cause damage to the seed coat and food reserves.

The significantly variable changes in the shoot, root length, and fresh and dry weights of *J. procera* seedlings treated with the three investigated growth regulators are recorded in Table 1. The highest shoot length was measured in seedlings treated with 100 ppm IAA (3.6 cm), followed by 50 ppm IAA (3.4 cm), 50 ppm GA$_3$ (3.1 cm), 100 ppm GA$_3$ (2.9 cm), 50 ppm kinetin (2.9 cm), and 100 ppm kinetin (2.8 cm). Moreover, pretreatment of juniper seeds with either sulfuric or acetic acid induced the lowest increase in shoot, root length, and fresh and dry weights compared with other treatments. The most pronounced growth parameters were recorded after IAA treatment. The significant increase in shoot and root lengths and fresh and dry weights of *J. procera* seedlings obtained in this study (Table 1) could presumably be ascribed to the positive effects of exogenously applied growth regulators on cell elongation, division, and extensibility (*Taiz & Zeiger, 2006*). The discrete changes in different growth attributes are hypothesized to be associated with or the result of discrete changes in the metabolic and hormonal activities of the developed seedlings. Thus, the appreciable increase recorded in the growth parameters of juniper seedlings in response to either relatively low (50 ppm) or high (100 ppm) concentrations of the investigated growth regulators could reflect relatively high measurable metabolic

activities that are responsible for successfully growing seedlings. The rate of metabolism in seeds during germination is controlled by signals originating from the embryo, which are mostly coherently related to its sink capacity (*Taiz & Zeiger, 2006*).

After 90 days of age, the root length was highest in IAA (100 ppm; 7.6 cm), with a seedling fresh weight of 0.19 g. The exogenous application of growth regulators may improve growth by increasing cell multiplication and elongation, resulting in rapid plant growth and development (*Anjanawe et al., 2013*). Consequently, the fresh and dry weights of juniper seedlings were significantly pronounced under IAA (50 and 100 ppm) and GA$_3$ (50 ppm) treatments, which might be because of mobilized water and the translocation of nutrients at higher levels, which stimulated the excess synthesis of photosynthetic products. Additionally, the stimulating effect of IAA may be attributed to cell enlargement and increased photosynthetic processes (*Naeem et al., 2004*), with cell division accompanied by augmentation of building units such as polysaccharides and total sugar content (*Sadak et al., 2013*). The results obtained in this study are in agreement with those of *Atteya et al. (2018)*, who confirmed the stimulating role of GA$_3$ on the growth parameters of *Simmondsia chinensis*.

## Effect of growth regulators on the photosynthetic pigments of *Juniperus procera* seedlings

Data in Table 2 show that the exogenous application of the three investigated phytohormones significantly increased the contents of chlorophylls a and b as well as carotenoids in the leaves of *J. procera*. After 90 days of application, the highest levels of chlorophylls a and b and carotenoids in *Juniperus communis* leaves were observed at 50 ppm GA$_3$, followed by 100 ppm GA$_3$. The value of chl a reached approximately 401.6 and 380.5 μg/g FW at 50 and 100 ppm GA$_3$, respectively (Table 2). Moreover, presoaking of *J. procera* seeds in either 90% H$_2$SO$_4$ or 20% CH$_3$COOH resulted in lower chl a and b and carotenoid contents of *J. procera* leaves compared with other investigated treatments. The lowest values of chl a and b and carotenoids were noticeable in the acetic acid treatment, which reached 172.1, 45.3, and 100.8 μg/g FW, respectively (Table 2).

The significant increase obtained in this study in fresh and dry weights of *J. procera* seedlings in response to relatively low or high concentrations of GA$_3$, IAA, and kinetin with a concomitant increase in chlorophyll a and b and carotenoid contents of leaves could indicate that these growth regulators may stimulate the photosynthetic efficiency of *J. procera* seedlings and improve the rate of assimilation in leaves with accumulation of photosynthates. A significant increase in photosynthesis has previously been described as an important response to growth regulators (*El Karamany, Sadak & Bakry, 2019*). The promoted effect of either GA$_3$, IAA, or kinetin on photosynthetic pigments is presumably due to the action of phytohormones as coenzymes in the metabolism of plants, which play a vital role in the formation and/or retardation of photosynthetic pigments (*Jacobs, 1979*). The increments in photosynthetic pigments in *J. procera* leaves in this study were consistent with the findings of 49 *Simmondsia chinensis*, which confirmed that growth regulators, particularly gibberellic acid, enhanced the production of photosynthetic pigments. Moreover, this increase in pigments in response to GA$_3$, IAA, and kinetin

**Table 3   Effect of pre-treatment with IAA at 50 ppm, IAA at 100 ppm, GA$_3$ at 50 ppm, GA$_3$ at 100 ppm, kinetin at 50 ppm kinetin at 100 ppm, 90% sulphuric acid and 20% acetic acids on total soluble sugars, proteins and oil % content of *juniper procera* seedling.** Data are means of three replications ± SE. Each value is the mean of three replicates ± SE.

| Treatment | Total soluble sugars (mg/g F.wt) | Total soluble proteins (mg/g F.wt) | Oil% |
|---|---|---|---|
| IAA (50 ppm) | 3.5 ± 0.12[c] | 7.3 ± 0.17[bc] | 40.8 ± 2.4[c] |
| IAA (100 ppm) | 5.6 ± 0.17[b] | 7.8 ± 0.11[b] | 55 ± 2.9[b] |
| GA3 (50 ppm) | 7.6 ± 0.05[a] | 10.5 ± 0.6[a] | 68.5 ± 0.3[a] |
| GA3 (100 ppm) | 6.9 ± 0.06[a] | 10.1 ± 0.05[a] | 58 ± 1.7[b] |
| Kinetin (50 ppm) | 5.1 ± 0.06[b] | 6.6 ± 0.06[cd] | 44.6 ± 1.2[c] |
| Kinetin (100 ppm) | 4.9 ± 0.5[b] | 6.5 ± 0.23[de] | 26.7 ± 1.8[c] |
| Sulphuric acid (90%) | 3.7 ± 0.1[c] | 5.8 ± 0.12[e] | 18.2 ± 0.12[e] |
| Acetic acids (20%) | 3.5 ± 0.28[c] | 4 ± 0.23[f] | 21.7 ± 0.6[e] |

**Notes.**

Columns with different lowercase letters are significantly different at $p < 0.05$.

was ascribed to their action in increasing the number and size of chloroplasts (*Ivanov & Angelov, 1997*; *Chen et al., 2003*; *Muthuchelian, Meenakshi & Nedunchezhian, 2003*). Similarly, the presoaking treatment of *J. procera* seeds with either GA$_3$, IAA, or kinetin at a relatively low or high concentration significantly increased all growth attributes (shoot and root lengths, fresh and dry weights, chlorophylls a and b, and carotenoids content, which indicate the role of these hormones to improve photosynthetic efficiency. The data in this study are in accordance with those of *Mousa, El-Sallami & Ali (2001)*, who found that the application of gibberellic acid increased the carotenoid content in *Nigella sativa* leaves compared with cytokinin application.

## Effect of growth regulators on total soluble sugar, protein, and oil content of *Juniperus procera* seedlings

Application of GA$_3$, IAA, or kinetin with relatively low (50 ppm) or high (100 ppm) concentrations caused a significant increase in total soluble sugar, protein, and oil content in *J. procera* seedlings compared with 90% sulfuric acid or 20% acetic acid (Table 3). The highest values of total soluble carbohydrate, protein, and oil content were verified in 50 ppm GA$_3$ 7.6, 10.5 mg/g FW, and 68.5%, respectively (Table 3).

The results of this study agree with many reports by *Ibrahim et al. (2007)*, *Sadak et al. (2013)*, and *Choudhury et al. (2013)* on faba bean and tomato plants, which confirmed the role of these hormones in the improvement of plant growth and development due to the excessive synthesis of Lemna minor starch (*Prajapati et al., 2015*). Moreover, an increase in the total soluble sugar level may be useful in producing vigorous plants. The positive role of sugar in plant growth and development is because it provides the energy required for the synthesis of new cells and cell division and enlargement. Additionally, pretreatments

**Table 4  Effect of pre-treatment with IAA at 50 ppm, IAA at 100 ppm, GA$_3$ at 50 ppm, GA$_3$ at 100 ppm, kinetin at 50 ppm, kinetin at 100 ppm, 90% sulphuric acid and 20% acetic acids on IAA, GA$_3$, kinetin and ABA content of _Juniper procera_ seedling.** Data are means of three replications ± SE. Each value is the mean of three replicates ± SE.

| Treatment | IAA (μg/100 g FW) | GA$_3$ (μg/100 g FW) | Kinetin (μg/100 g FW) | ABA (μg/100 g FW) |
|---|---|---|---|---|
| IAA (50 ppm) | 227.9 ± 0.06[a] | 1408.2 ± 1.2[c] | 346.3 ± 0.77[e] | 21.9 ± 0.6[c] |
| IAA (100 ppm) | 230.7 ± 2.9[a] | 1748.8 ± 1.2[c] | 255.9 ± 1.1[f] | 20.5 ± 0.3[c] |
| GA$_3$ (50 ppm) | 180.8 ± 0.57[b] | 1796.2 ± 1.2[b] | 378.8 ± 1.1[d] | 12.07 ± 0.02[f] |
| GA$_3$ (100 ppm) | 164.3 ± 2.3[c] | 1875.1 ± 1.1[a] | 445.4 ± 3.0[c] | 14.1 ± 0.6[f] |
| Kinetin (50 ppm) | 147.1 ± 1.2[c] | 1225.6 ± 1.3[f] | 495.7 ± 2.8[b] | 17.07 ± 0.5[e] |
| Kinetin (100 ppm) | 138.0 ± 1.7[e] | 1300.8 ± 2.9[e] | 576.7 ± 0.12[a] | 25.7 ± 1.8[c] |
| Sulphuric acid (90%) | 54.3 ± 0.17[f] | 799.8 ± 0.6[h] | 129.8 ± 0.63[g] | 30.9 ± 1.2[b] |
| Acetic acids (20%) | 43.1 ± 1.2[g] | 817.8 ± 1.2[g] | 128.1 ± 0.06[g] | 38.6 ± 1.4[a] |

**Notes.**

Columns with different lowercase letters are significantly different at $p < 0.05$.

with phytohormones generally enhanced the total soluble carbohydrate content, which was mainly accompanied by enhanced chlorophyll content (_Alsokari, 2011_).

The cytokinin-stimulating effect may be related to protein synthesis because of the postulation that their activity is associated with putative receptor systems of specific proteins located in the nucleus and cytosol. Additionally, numerous studies have revealed that IAA augmented the levels of nitrogenous compounds, primarily total soluble proteins, in many plant types (_Sarkar, Haque & Karim, 2002_; _Prajapati et al., 2015_). Moreover, it was noticeable that the total soluble protein content in _J. procera_ seedlings increased significantly with GA$_3$ application. In our study, the increase in these primary metabolites by pretreatments with GA$_3$, IAA, and kinetin could have contributed to the acceleration of the photosynthesis process because it was associated with an increase in photosynthetic pigments (Table 2).

## Effect of growth regulators on endogenous phytohormones in _Juniperus procera_ seedlings

At the hormonal level, _J. procera_ seedlings resulted from presoaked seeds in the relatively low and high concentrations of the three investigated growth regulators, which induced a marked increase in their IAA, GA$_3$, and cytokinin contents, whereas ABA content was decreased compared with $H_2SO_4$ or $CH_3COOH$ presoaked seedlings (Table 4).

Such hormonal imbalance is likely to impair the process of seed germination either directly _via_ interference with their role in the growth requirements of cell elongation, expansion, _etc._, or indirectly by affecting the metabolic machinery of the seedlings, as growth hormone-inducing growth requires continuous metabolic input.

Auxins are implicated in the stimulation of cell elongation by increasing cell wall extensibility or plasticity, which is mediated by stimulating H+-ATPase, which is concomitant with proton extrusion and loosening of cell wall cellulose fiber, synthesis of expansions, *etc.* (*Taiz & Zeiger, 2006*). The significant increase in the endogenous content of auxins in response to the relatively low or high concentrations of phytohormones used in this investigation resulted in enhanced growth of seedlings expressed in terms of a significant increase in the lengths of shoots and roots and fresh and dry weights of *J. procera* seedlings (Table 1). The most pronounced increments in IAA, $GA_3$, and kinetin levels were recorded in 100 ppm IAA-, 100 ppm $GA_3$-, and 100 ppm kinetin-soaking seedlings.

Additionally, $GA_3$ plays an important role in seed germination and growth. Low $GA_3$ content affects metabolic activities *via* certain mechanisms, which could contribute to retardation of the efficiency of maize in the conversion of fats into sugars *via* the glyoxylate cycle because $GA_3$ plays a crucial role in this pathway (*Heldt, Piechulla & Heldt, 2011*). The role of $GA_3$ in the synthesis and/or stimulation of hydrolases in the mobilization of seed reserves has been established (*Jacobsen, Gubler & Chandler, 1995*).

The effective role of either $GA_3$ or cytokinins in the use of reserves, a prerequisite step for providing respiratory substrates, is well established. Therefore, the accumulation of endogenous $GA_3$ and cytokinins is expected to increase the respiratory intensity of *J. procera* seedlings. Consequently, an observed increase in respiration, a sharp increment in the growth requirements of energy and different building blocks involved in different growth aspects. Such increases in different endogenous growth bioregulatory pathways could improve their biosynthesis and/or decrease their degradation (*Taiz & Zeiger, 2006*). $GA_3$ initiates proteolytic enzymes that release tryptophan, a common precursor of IAA (*El-Saeid, Abou-Hussein & El-Tohamy, 2010*). An increase in endogenous hormones is associated with the generation of some metabolites, particularly sugar (*Shah, 2011*). Moreover, high $GA_3$ levels induced an increase in the availability of water uptake, which led to a rapid increase in the fresh weight of seedlings. Furthermore, kinetin-soaking treatment elevated cytokinin levels in *J. procera* seedlings, leading to an increase in seed division, enhanced seed germination, and improved growth performance (*Vamil, Haq & Agnihotri, 2010*).

The marked decrease in the ABA content of *J. procera* seedlings in response to relatively low or high concentrations of $GA_3$, IAA, or kinetin plays a role in improving growth because ABA interferes with the mode of action of $GA_3$ and suppresses $GA_3$-mediated genes encoding hydrolytic enzymes in germinating seeds and seedlings (*Walker-Simmons et al., 2000*). Moreover, ABA, which appears to delay DNA replication and inhibit DNA metabolism, inhibits seed germination and plant growth (*Vazquez-Ramos, 1993*). The data of this study are in agreement with those obtained by *El-Saeid, Abou-Hussein & El-Tohamy (2010)*, *Barakat Nasser (2011)*, and *Sadak et al. (2013)* on cowpea, wheat, and faba bean plants.

The increase in endogenous phytohormone levels due to the application of the three hormones under investigation may be attributed to a reduction in the activity of their degrading enzymes (*Soda et al., 2021*). Meanwhile, the decrease in the ABA content observed in *J. procera* seedlings treated with the investigated hormones could be attributed

to the transfer of isopentenyl pyrophosphate, which is the main precursor of cytokinin and/or $GA_3$ biosynthesis, in place of ABA (*Hopkins & Hünter, 2004*).

## CONCLUSIONS

The results of this study revealed that morphological and biochemical evidence supports the bioactive roles of three different growth regulators (IAA, $GA_3$, and kinetin) and 90% sulfuric acid and 20% acetic acid in breaking seed dormancy and improving seedling growth of *J. procera*, which is facing a notable decline in its population worldwide. The three phytohormones succeeded in breaking the dormancy of *Juniperus* seeds, which may be attributed to their role in stimulating seed germination and seedling growth. The highest percentage of germination was recorded in $GA_3$ (50 ppm) treatment, which has a crucial role in seed germination. Additionally, chemical treatments could break *Juniperus* seed dormancy; the lowest value was recorded in acetic acid followed by sulfuric acid. Although this acid helps in breaking the seed coat, it also has a damaging effect on the embryo. Conversely, IAA (100 ppm) had the best results for improving seedling growth and was associated with an increase in photosynthetic pigments, total soluble sugars, proteins, and percentage of oil.

### Funding
The authors received no funding for this work.

### Competing Interests
The authors declare there are no competing interests.

### Author Contributions
- Alae Ahmad Jabbour conceived and designed the experiments, performed the experiments, analyzed the data, authored or reviewed drafts of the article, and approved the final draft.
- Abdulaziz Alzahrani performed the experiments, analyzed the data, prepared figures and/or tables, and approved the final draft.

### Data Availability
The raw data are available in the Supplemental File.

### Supplemental Information
Supplemental information for this article can be found online at http://dx.doi.org/10.7717/peerj.17236#supplemental-information.

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
