# Peer review of "The impact of chemical and hormonal treatments to improve seed germination and seedling growth of Juniperus procera Hochst. ex Endi"

_PeerJ, doi:10.7717/peerj.17236_

## Round 0.1 · original submission · Minor Revisions

Dear Authors,
The reviewers agree with publication but suggest minor revisions.
I invited the authors to make all needed changes and respond with point-to-point letter to reviewers in due time.
Best regards,
C. Arena

**Language Note:** The review process has identified that the English language must be improved. PeerJ can provide language editing services - please contact us at [email protected] for pricing (be sure to provide your manuscript number and title). Alternatively, you should make your own arrangements to improve the language quality and provide details in your response letter. – PeerJ Staff

·

Basic reporting

The present manuscript titled as “The impact of chemical and hormonal treatments to improve seed germination and seedling growth of Juniperus procera Hochst. ex Endi.”, assesses the effect of chemical and hormonal treatments in improving the seed germination and seedling growth of Juniper the seeds either subjected to chemical scarification with 90% sulfuric acid and 20% acetic acid for 6min or hormonal treatment by seed-soaking in two concentrations (50 & 100 ppm) of three growth regulators namely, indole acetic acid (IAA), gibberellins (GA3) and kinetin (k) for 72. The idea and concept of the study is interesting and considered as one of the strategies for solving the problem faced Juniperus plant.
The manuscript is well-written but still needs some minor changes.

Experimental design

Abstract:
The abstract background needs to connect with the hypothesis and then the objective, methods, results, and conclusion.
The experimental design is satisfactory and the methods describe sufficient details, but still needs some minor changes.
Provide the Geographical coordinates for Baljurashi, Al-Baha, Saudi Arabia, site.
Line 107: correct seeds instead of seed.
Line 109:The authors mentioned 10 replicates for the of growth measurement, but in Table 1, the legend mentioned three replicates were used. These are contradictory. How the authors will explain this?
Line 135: Check the plant name (common juniper) or Juniperus procera

Validity of the findings

Results:
The results section is well written but still needs some minor changes
The significant letters must be superscript in all tables.
Discussion
The discussion section is well written.
Conclusion
The conclusion section is well-written and sufficient.
References
Please unify the style according to the journal instructions.

Additional comments

no comments

Reviewer 2 ·

Basic reporting

The abstract presents a comprehensive overview of a research study aimed at addressing the challenges of seed dormancy and natural regeneration decline in Juniperus procera populations in Saudi Arabia. The study explores the effects of chemical and hormonal treatments on seed germination and seedling growth, providing valuable insights into potential strategies for mitigating population decline.
Strengths:
1. Clear Problem Statement:
The abstract effectively highlights the specific challenges faced by Juniperus procera populations in Saudi Arabia, emphasizing seed dormancy and natural regeneration decline as the primary contributors to the species' decline.
2. Methodological Clarity:
The methods employed in the study are clearly outlined, including details on the chemical scarification and hormonal treatments. The inclusion of a control group adds rigor to the experimental design.
3. Data Presentation:
The abstract presents key findings on seed germination and seedling growth, pointing out the notable effects of different treatments. The emphasis on biochemical parameters such as photosynthetic pigments, soluble sugars, proteins, and phytohormone contents adds depth to the study.
4. Relevance to Conservation:
The study addresses a significant ecological concern by proposing practical solutions to enhance seed germination and seedling growth. The potential applications of the findings for conservation efforts are well-stated.
Suggestions for Improvement:
1. Additional Context:
While the abstract provides a clear overview, a brief introduction providing additional context on the ecological importance of Juniperus procera in Saudi Arabia and the broader implications of its decline could enhance the reader's understanding.
2. Statistical Information:
The abstract lacks information on statistical analyses employed in the study. Including key statistical measures (e.g., mean, standard deviation) for seed germination and seedling growth could strengthen the scientific rigor of the research.
3. Ecological Implications:
The abstract briefly mentions the decrease in abscisic acid (ABA) content compared to chemical treatments. Expanding on the ecological implications of this finding and discussing potential long-term effects on the species or ecosystem would enrich the conclusion.

Intro & background to show context and the literature well referenced & relevant. The figures are relevant, high quality, well labelled & described.
Experimental Design, Material, and Methodology
Strengths:

Overall Impression:
The paper effectively communicates the research objectives, methods, and key findings of the study. With slight improvements in contextualization, statistical reporting, and ecological implications, this research holds promise for contributing valuable insights into the conservation of Juniperus procera in Saudi Arabia.

Experimental design

The paper exhibits a robust experimental design, meticulous seed collection, and precise methodologies. Thorough sampling, including at least 10 randomly selected mother trees for seed collection, showcases a strong foundation. The use of sodium hypochlorite for sterilization, division into treatment groups, and inclusion of both chemical and hormonal treatments adds depth to the study. The germination experiment, with three replicates and clear criteria for assessment, demonstrates a thoughtful approach. The seedling growth experiment, employing a randomized complete block design and careful management of environmental conditions, contributes to the reliability of the results.
Suggestions for Improvement:
The paper could benefit from explicit reporting of statistical analyses to enhance scientific rigor. Including ecological context on Juniperus procera in Saudi Arabia would provide a more comprehensive background. Consider providing supplementary data or details on replicate variability for increased transparency.
The study shows promise in addressing seed dormancy in Juniperus procera. The detailed experimental design and methodologies contribute to the paper's overall strength. Addressing the suggested improvements would further elevate the scientific rigor and impact of the research.

Validity of the findings

Strengths:

Thorough Exploration:
The study thoroughly investigates the impact of growth regulators and chemical treatments on Juniperus procera seed germination and seedling growth, presenting a detailed examination of both morphological and biochemical evidence.

Phytohormone Efficacy:
The conclusion that IAA, GA3, and kinetin effectively break seed dormancy is well-founded, with a particularly strong result for GA3 (50 ppm). The emphasis on the known roles of these hormones in seed processes adds credibility.

Chemical Treatment Insight:
The differentiation between sulfuric acid and acetic acid effects on seed germination, with a nuanced explanation for the lower germination percentage with acetic acid, demonstrates a keen understanding of the potential damage to the embryo and enriches the interpretation of results.

Biochemical Parameters and Seedling Growth:
The identification of IAA (100 ppm) as the most effective treatment for seedling growth, supported by increases in photosynthetic pigments, total soluble sugars, proteins, and oil percentage, aligns with established knowledge and enhances the depth of the study.

Constructive Suggestions:

Statistical Rigor:
Strengthening the paper's validity could be achieved through a more detailed presentation of statistical analyses. Including measures of variability and significance would provide readers with a clearer understanding of the robustness of the reported effects.

Ecological Context:
The addition of a brief discussion on the broader ecological context of Juniperus procera would elevate the study. Considering potential long-term effects on natural populations and ecosystems would enhance the ecological relevance of the findings.

Seedling Viability Consideration:
The paper could benefit from a discussion on the viability and health of the seedlings resulting from the treatments. Assessing the long-term survival and growth of the seedlings would contribute valuable insights into the practical implications of the study.

Overall Impression:
The study is a commendable exploration of strategies to address seed dormancy in Juniperus procera. While the findings are robust, addressing the suggested improvements, especially in statistical reporting and ecological considerations, would further elevate the scientific merit of the research. The study adds valuable insights to the understanding of seed germination and seedling growth in the context of population decline in Juniperus procera.

Reviewer 3 ·

Basic reporting

The preface is complete, the writing is clear, and the language and references are further improved.

Experimental design

This research project has a certain degree of originality and is useful for analyzing Juniperus procera Hochst Ex Endi. Germination and growth have certain novelty and potential applications. The writing of this research method is clear and the analysis is reasonable. However, a more careful analysis is necessary. Simple handling and changing trends lack certain significance.

Validity of the findings

This study analyzed the germination effects of seeds treated with different hormones and presented corresponding results. However, more detailed analysis of changes and more vivid presentation of results (such as in the form of images) will more effectively indicate the results.

---

## Round 0.2 · Minor Revisions

Dear Authors,
The manuscript has been consistently improved after the second round of revisions.

Please revise for English writing.

E.g.:
In the title!
In the Abstract: the Conclusion is too short and without a message.
"Conclusions
The present results revealed that, morphological and biochemical evidence supportive the bioactive roles of three different growth regulator"

Note: '&' should only be used for persons, not as a replacement of 'and'.

Please provide evidence for the proofreading by a fluent speaker/professional service.

Best Regards

---

## Round 0.3 · accepted · Accept

Dear Authors

The paper has been consistently improved after revision and it is now worthy of publication in PeerJ Life and Environment

The Section Editor noted:

> Scientifically, the paper is acceptable. However, I would recommend that the author polish the writing. For example, in Conclusions, 'The results of this study revealed that ' can be completely removed, which would improve readability. Also, two messages are mixed in the first sentence: (1) the decline of the population and 2) the results of the study). "The results of this study revealed that morphological and biochemical evidence supports the 344 bioactive roles of three different growth regulators (IAA, GA3, and kinetin) and 90% sulfuric acid 345 and 20% acetic acid in breaking seed dormancy and improving seedling growth of J. procera, 346 which is facing a notable decline in its population worldwide. "

·

Basic reporting

The present manuscript titled as “The impact of chemical and hormonal treatments to improve seed germination and seedling growth of Juniperus procera Hochst. ex Endi.”, assesses the effect of chemical and hormonal treatments in improving the seed germination and seedling growth of Juniper the seeds either subjected to chemical scarification with 90% sulfuric acid and 20% acetic acid for 6min or hormonal treatment by seed-soaking in two concentrations (50 & 100 ppm) of three growth regulators namely, indole acetic acid (IAA), gibberellins (GA3) and kinetin (k) for 72. The idea and concept of the study are interesting and considered as one of the strategies for solving the problem faced juniperus plant.
The manuscript is now in a better form and ready for publication. So, my decision to accept the manuscript to publish in Peerj Journal.

Experimental design

no comments

Validity of the findings

no comments

Additional comments

no comments

Reviewer 2 ·

Basic reporting

no comment

Experimental design

no comment

Validity of the findings

no comment

Additional comments

no comment